# Relationship Between the Sagittal and Coronary Alignment of the Tibiofemoral Joint and the Medial Meniscus Extrusion in Knee Osteoarthritis

**DOI:** 10.3390/healthcare12232412

**Published:** 2024-12-01

**Authors:** Masahiro Ikezu, Shintarou Kudo, Ryuzi Mitsumori, Wataru Iseki, Masato Shibukawa, Yasuhiko Iizuka, Masahiro Tsutsumi, Hidetoshi Hayashi

**Affiliations:** 1Inclusive Medical Sciences Research Institute, Morinomiya University of Medical Sciences, Osaka 559-8611, Osaka, Japan; ikezu1201@gmail.com (M.I.); masahiro_tsutsumi@morinomiya-u.ac.jp (M.T.); 2AR-Ex Medical Research Center, Setagaya 158-0082, Tokyo, Japan; 3Ueda Orthopedic Clinic, Ueda 386-0018, Nagano, Japan; mitsumori-valerianoryuji@ar-ex.jp (R.M.); shibukawa-masato@ar-ex.jp (M.S.); 4AR-Ex Spine Clinic, Setagaya 158-0082, Tokyo, Japan; iseki-wataru@ar-ex.jp; 5AR-Ex Oyamadai Orthopedic Clinic Tokyo Arthroscopy Center, Setagaya 158-0082, Tokyo, Japan; iizuka-yasuhiko@ar-ex.jp (Y.I.); hayashi-hidetoshi@ar-ex.jp (H.H.)

**Keywords:** knee osteoarthritis, medial meniscus extrusion, knee alignment, ultrasound imaging

## Abstract

**Background/Objectives**: This study aims to clarify the reproducibility, validity, and accuracy of tibial external-rotation alignment evaluation using ultrasound imaging and to investigate the relationship between medial meniscus extrusion (MME) and tibiofemoral alignment in both the sagittal and coronal planes in knee osteoarthritis (OA). **Methods**: Study 1 included 10 healthy participants. The tibial external-rotation angle was calculated using MRI. In the ultrasound imaging evaluation, the differences in the distance from the most posterior points of the tibial and femoral condyles to the skin were calculated as the medial and lateral condyle gaps, respectively. The mediolateral (ML) gap was calculated by subtracting the lateral condyle gap from the medial condyle gap. Study 2 included 63 patients with unilateral OA and 16 healthy controls. MME was compared according to the severity of OA, the degree of tibial rotation, and the presence or absence of a tibial posterior shift. **Results**: Ultrasound imaging examinations showed high intra- and inter-rater reliabilities (0.786–0.979). The ML gap significantly affected the tibial external-rotation angle, determined using MRI. The ML gap of ultrasound imaging was significantly correlated with the ML gap of MRI. MME was significantly higher in the Early OA group than in the Control group. There was no significant difference in MME based on the tibial rotation degree. The group with a tibial posterior shift exhibited significantly more MME than that without a posterior shift. **Conclusions**: Ultrasound imaging is useful for evaluating knee alignment. MME was found to be associated with the tibial posterior shift.

## 1. Introduction

Medial meniscal extrusion (MME) is a known risk factor for the progression of knee osteoarthritis (OA), partial or total knee replacement, or the surgical repair of a medial meniscus posterior root tear (MMPRT), which is defined as an avulsion injury or radial tear occurring at the posterior bony attachment [1,2]. In contrast, Makeiv et al. [3] hypothesized that MMPRT is not the sole contributing factor predisposing to MME; therefore, to prevent its progression, a better understanding of the mechanism of MME is needed to develop its treatment.

The mechanisms of MME include damage to the meniscus, such as posterior root and radial tears [4]. As for other mechanisms, MME is associated with medial tibial osteophytes and mechanical factors, such as the knee adduction moment (KAM) [5,6]. The KAM is also strongly related to the varus alignment of the knee: MME increases as knee varus alignment progresses, and the KAM increases it [7]. Although the knee varus is the alignment of the knee joint in the frontal plane, the effects of anteroposterior alignment in the sagittal plane and rotational alignment in the coronal plane have not been fully investigated.

The alignment of the tibiofemoral joint has been extensively studied through computed tomography [8] and magnetic resonance imaging (MRI) [9]. However, challenges such as radiation exposure and procedural simplicity persist in clinical practice. The quadriceps angle (Q angle), influenced by femoral anteversion and tibiofemoral angle [10], exhibits variable reproducibility [11]. Therefore, we have developed a novel method utilizing ultrasound imaging to assess the alignment of the tibiofemoral joint in both the sagittal and coronal planes. The medial collateral ligament and semimembranosus tendons provide tension to the medial meniscus [12]. These structures undergo increased tension during the external rotation of the lower-leg knee extension. Investigating the relationship between MME and the alignment of the tibiofemoral joint may thus provide useful information to clarify the mechanism of MME.

The primary purpose of this study was to clarify the reproducibility, validity, and accuracy of tibial-rotation alignment evaluation using ultrasound imaging (Study 1). The secondary purpose of this study was to investigate the relationship between MME and tibiofemoral alignment in both the sagittal and coronal planes in knee osteoarthritis (Study 2). We hypothesized that MME occurs during internal tibial rotation or posterior tibial translation relative to the femur.

## 2. Materials and Methods

### 2.1. Study 1

#### 2.1.1. Subjects

The subjects were 10 healthy adults with 20 knees (10 males; mean age, 27.2 ± 3.7 years; height, 174.6 ± 5.0 cm; and weight, 66.2 ± 7.7 kg). The inclusion criterion was no prior or current knee disease, including diseases affecting the bone, cartilage, ligament, tendon, joint capsule, or meniscus. The exclusion criterion was individuals who had difficulty maintaining the required posture during the measurement.

This study was conducted after obtaining ethical review approval from the Morinomiya University of Health Sciences (No. 2021-083). In accordance with the Declaration of Helsinki, the rights of the subjects were explained, and their consent was obtained on paper. Prior and current knee diseases were obtained from oral history and medical records. Our data collection started on 5 October 2021 and ended on 8 February 2022.

#### 2.1.2. Ultrasound Imaging Examinations

Ultrasound imaging examinations were performed using a SONIMAGE HS1 (KONICA MINOLTA, Tokyo, Japan) with an 18 MHz linear transducer in B-mode. Measurements were performed with the patients in the prone position on a bed, with the knee joint in full extension and the center of the patella positioned at the edge of the bed. A physical therapist performed the ultrasound imaging measurements. The probe was placed at the popliteal fossa, and the most prominent posterior points of the medial femoral and tibial condyles and the most prominent posterior points of the lateral femoral and tibial condyles were visualized on the long-axis images (Figure 1). The medial condyle gap was calculated as the difference between the distance from the most prominent posterior medial tibial condyle to the skin and the distance from the most prominent posterior medial femoral condyle to the skin. The lateral condyle gap was calculated as the difference between the distance from the most prominent posterior lateral tibial condyle to the skin and that from the most prominent posterior lateral femoral condyle to the skin. The mediolateral (ML) gap was calculated by subtracting the lateral condyle gap from the medial condyle gap. The ML gap was assigned a positive value for external rotation and a negative value for internal rotation.

#### 2.1.3. MRI Examinations

All MRI examinations were performed using a 1.5T system (Oval; HITACHI, Chiba, Japan), with the extremity coils being used as receiver coils. The knee joint was selected as the imaging site for this study. Measurements were performed in the prone position with the knee joint in full extension, and the center of the patella was positioned at the center of the receiver coil. The MRI measurements were performed by a single radiologist. The tibial external-rotation angle was evaluated using axial images, as described by Vassalou et al. [13] (Figure 2). The femoral rotation angle was defined as the angle between the horizontal line and the line of the subchondral bone of the posterior femoral condyle at the most prominent point [13]. The tibial rotation angle was defined as the angle between the horizontal line and the line of the posterior cortical margin of the tibial condyles using one slice above the head of the fibula [13]. The tibial external-rotation angle was calculated by subtracting the femoral rotation angle from the tibial rotation angle; it was assigned a positive value for the external rotation and a negative value for the internal rotation.

The ML gap of MRI was evaluated using sagittal images. The ML gap was based on ultrasound imaging examinations, and the distance between the posterior margins of the femoral and tibial medial condyles was the medial condyle gap. The distance between the posterior margins of the femoral and tibial lateral condyles was measured as the lateral condyle gap. The ML gap was calculated by subtracting the lateral condyle gap from the medial condyle gap.

#### 2.1.4. Q-Angle Examinations

The Q angle was measured using a standard goniometer. The measurements were performed with the patient supine on a bed with the knee joint in full extension. The Q angle was defined as the angle formed by a line from the superior anterior iliac spine to the center of the patella and a line from the center of the patella to the tibial tuberosity [10]. The angle was measured in degrees (°).

#### 2.1.5. Intra-Rater Reliability and Inter-Rater Reliability

The subjects were five healthy adults with 10 knees (five males; mean age, 26.6 ± 3.1 years; height, 175.6 ± 5.6 cm; and weight, 65.8 ± 8.5 kg) and without a history of knee joint disease. Ultrasound imaging examinations were performed in the same manner as the US measurement method described above. Retests were performed at intervals of three to seven days. The inter-rater reliability of the two investigators was also confirmed in the same five healthy adults with 10 knees.

#### 2.1.6. Statistical Analysis

The reproducibility of the ultrasound imaging examinations was assessed using the intraclass correlation coefficient (ICC). The intra- (ICC 1, 3) and inter-rater reliability (ICC 2, 3) were calculated. The criteria for ICC were as follows: <0.00 = poor, 0.00–0.20 = slight, 0.21–0.40 = fair, 0.41–0.60 = moderate, 0.61–0.80 = substantial, and 0.81–1.00 = almost perfect [14]. Minimal detectable change at the 95% CI (MDC95%) was calculated as follows: MDC95% = 1.96 × √2 × SEM [15]. SEM refers to the standard error of measurement and is calculated as SEM = SD√(1 √ICC) [15]. Single and multiple regression analyses were performed with the MRI tibial external-rotation angle as the dependent variable and the ML gap of ultrasound imaging and Q angle as the independent variables. The statistical correlation between the ultrasound imaging and the MRI ML gaps was analyzed using Spearman’s correlation. SPSS Statistics 27 (IBM Corp., Armonk, NY, USA) was used for the statistical analysis.

### 2.2. Study 2

#### 2.2.1. Subjects

Sixty-three patients with unilateral knee OA (18 males and 45 females) who had been diagnosed with knee OA by an orthopedic surgeon were included in this study. The Control group consisted of 16 healthy subjects with 32 knees (6 males and 10 females) whose average age, height, and weight matched those of the knee OA group. The inclusion criteria for the knee OA group comprised individuals aged 50 years or older experiencing pain from the anterior to medial side of the knee joint during loading, those with available radiographic images, and those deemed suitable for nonoperative treatment. The exclusion criteria were rheumatoid arthritis, OA secondary to trauma, previous knee surgery, severe deformities that made ultrasound imaging difficult, and neurological disorders.

This study was conducted after obtaining ethical review approval from the Morinomiya University of Health Sciences (No. 2022-095). In accordance with the Declaration of Helsinki, the rights of the subjects were explained, and their consent was obtained on paper. Radiographic images and symptoms were obtained from oral history and medical records. Our data collection started on 5 November 2022 and ended on 25 February 2023. All data collection was performed at an orthopedic clinic.

#### 2.2.2. Procedures

In the knee OA group, knee deformities were classified by Kellgren and Lawrence (KL) grades 0–4 by a single orthopedic surgeon. KL grades 0 and 1 were defined as early OA and KL grades 2 or higher were deemed modulate OA [16].

The knee joint extension angle was measured in five-degree increments by using a goniometer, and the presence of limitations was recorded as a negative sign.

Ultrasound imaging was performed using a Noblus device (Hitachi Medical Systems, Tokyo, Japan) with an 18 MHz linear transducer in B-mode. MME was performed with the knee extended as much as possible, and the midportion of the medial meniscus was captured as a long-axis scan of the MCL. The vertical distance from the line segment connecting the medial end of the medial condyle of the femur and tibia, excluding the osteophytes, to the outer edge of the medial meniscus was calculated (Figure 3). Articular swelling was defined as a hypoechoic area at the suprapatellar porch, and joint effusion was measured as the maximum distance between the superficial and deep synovium.

Tibiofemoral joint alignment was based on ultrasound imaging examinations in Study 1. The distance between the posterior margins of the femoral and tibial medial condyles was deemed the medial condyle gap, and the distance between the posterior margins of the femoral and tibial lateral condyles was measured as the lateral condyle gap (Figure 1). The ML gap was calculated by subtracting the lateral condyle gap from the medial condyle gap.

#### 2.2.3. Statistical Analysis

Age, height, weight, MME, joint effusion, medial condyle gap, lateral condyle gap, and tibial external-rotation alignment were compared among the three groups (Control, Early OA, and Moderate OA) using one-way ANOVA. The tibial external-rotation alignment was calculated based on the regression equation obtained in Study 1. The post hoc test was performed using the Bonferroni method with a significance level of less than 5% for comparisons among the three groups. The mean and standard deviation of the medial condyle gap, lateral condyle gap, and tibial external-rotation alignment of the Control group were used to define the standard distribution as within one standard deviation, and the tibial external-rotation alignment of the OA group was defined as being within the standard distribution. The OA group was further classified into three groups: normal rotation, excessive external rotation, and excessive internal rotation. Additionally, age, height, weight, MME, extension angle, and joint effusion were compared among the three groups using one-way ANOVA. The post hoc test was performed using the Bonferroni method with a significance level of less than 5% for comparisons among the three groups. The tibial posterior shift was defined as a case in which either or both the medial and lateral condyle gaps were less than the standard distribution of the control. The OA group was divided into two subgroups according to the absence or presence of posterior shift, and the results for age, height, weight, MME, extension angle, joint effusion, and tibial external-rotation alignment were compared between patients with and without posterior shift using a *t*-test. SPSS Statistics 27 (IBM Corp., Armonk, NY, USA) was used for the statistical analysis.

## 3. Results

### 3.1. Study 1

The intra- (ICC 1,3) and inter-rater reliabilities (ICC 2, 3) for the ultrasound imaging examinations were from 0.786 to 0.979 (Table 1). In this study, the measurement of the ultrasound imaging showed substantial to almost perfect reliability according to the criteria of Landis and Koch [14].

The results of the single regression analysis showed that the ML gap of ultrasound imaging significantly affected the tibial external-rotation angle determined via MRI (*p* < 0.01; R^2^ = 0.755). The Q angle significantly affected the tibial external-rotation angle determined via MRI (*p* = 0.04; R^2^ = 0.211). The results of the multiple regression analysis are presented in Table 2. The ML gap of ultrasound imaging significantly affected the tibial external-rotation angle determined via MRI (*p* < 0.01; R^2^ = 0.764). The Q angle did not significantly affect the tibial external-rotation angle determined via MRI (*p* = 0.45). The regression equation for calculating the tibial external-rotation angle was as follows (1):Tibia external-rotation alignment (degrees) = 1.095 × (ML gap) + 1.732(1)

The ML gap of the ultrasound imaging was found to be significantly and highly correlated with the ML gap of the MRI (Figure 4; r = 0.78, *p* < 0.01).

### 3.2. Study 2

In Study 2, 63 knees of 63 patients with knee OA participated in the experiment, and 31 knees of 16 healthy subjects matched the age of the knee OA group without knee pain. One healthy knee had a history of surgery and was excluded from the study.

The parameters of the Healthy, Early OA, and Moderate OA groups based on the KL classification of the subjects are shown in Table 3. Significant differences were observed in age, height, MME, knee extension range of motion, and joint effusion; the Early OA group was significantly younger than the Control group. The height of the Early OA group was significantly lower than that of the Control group. The MME was significantly higher in the Early OA group than in the Control group. In addition, the knee extension range of motion and joint edema were significantly different among all groups, and the more severe the KL classification, the more limited the extension range of motion and the greater the joint effusion.

The parameters of the normal rotation, excessive external rotation, and excessive internal rotation groups in the OA group according to the tibial external-rotation alignment in the Control group are shown in Table 4. The limitation of the extension range of motion was significantly lower in the excessive internal rotation group than in the normal rotation group. Moreover, joint effusion was significantly increased compared to that in the normal rotation group. The medial gap was significantly different among all groups and increased in the order of excessive internal rotation, normal rotation, and excessive external rotation. The lateral gap was significantly lower in the excessive internal rotation group than in the other two groups. However, there was no significant difference in MME among the three groups.

On the other hand, 24 knees were defined as knee OA with a posterior shift of less than 3.8 mm of the medial condyle gap and less than 4.6 mm of the lateral condyle gap; the other 39 knees were defined as knee OA without a posterior shift (Figure 5). The MME of the posteriorly shifted group, in which the tibia was shifted posteriorly compared to the medial and lateral gaps of the Control group, was significantly greater than that of the non-posteriorly shifted group (Table 5). In addition, the posteriorly shifted group showed a significantly lower knee extension angle and significantly greater joint effusion than the non-posteriorly shifted group.

## 4. Discussion

Study 1 aimed to elucidate the reproducibility, validity, and accuracy of tibial-rotation alignment evaluation using ultrasound imaging. Our findings indicate that the assessment of tibial rotation alignment through ultrasound imaging exhibits high levels of reproducibility, validity, and accuracy. This study demonstrates the feasibility of assessing tibial rotation alignment in absolute values using an ultrasound imaging system. Integration with an inertial measurement unit and optical motion analysis could facilitate the exploration of the previously unknown relationship between tibial rotation motion and associated symptoms.

Study 2 aimed to investigate the relationship between MME and tibiofemoral alignment in both the sagittal and coronal planes in knee osteoarthritis. Tibial rotation alignment was not found to be associated with MME. The MME of the posteriorly shifted group was significantly greater than that of the non-posteriorly shifted group. These results suggest that tibial posterior shift alignment may be associated with MME.

The hypothesis of Study 2 was that the excessively externally rotated knee would demonstrate a greater MME. However, no relationship between the rotational alignment of the tibia and MME was found. MME leads to decreased tibial plateau coverage, increasing cartilage load-bearing and contributing to the progression of OA [1,4]. Wang et al. [17], in a retrospective review of 131 patients who underwent arthroscopic surgery for knee OA and were followed up for 4 years, found that arthroscopic surgery was also beneficial in patients with major MME (>3 mm) in terms of pain relief. In contrast, the cases in which the tibia was positioned posteriorly to the femur had greater MME. In addition, the cases in the posteriorly shifted group showed greater limitations of extension range of motion and joint effusion than those in the non-posteriorly shifted group. Naraedo et al. [18] reported that knee effusion and MME were associated with pain in knee OA. In other words, the abnormal sagittal alignment of the tibiofemoral joint is a risk factor for the progression of knee OA.

Increased varus alignment is a typical alignment problem in knee osteoarthritis [7]. MME during gait is associated with decrease in knee rotation in early OA [19]. However, rotational alignment remains controversial, and Ikuta et al. [20] reported no relationship between KL severity and internal/external rotation alignment. In contrast, Nagao et al. [21] found that external rotation at maximal knee extension and rotation of the screw home movement proportionately decreased in knee osteoarthritis. Matsui et al. [22] reported that the tibia was externally rotated in medial OA, and Yagi et al. [23] found no change compared with healthy subjects. In other words, a certain view of the lower-leg rotation alignment was not obtained. By contrast, other studies have shown that the tibia moves posteriorly with changes in the sagittal plane [20,24]. In this study, the lower leg was rotated internally and externally in some cases. The relationship between rotational alignment and the MME is also unclear. However, we found that knee OA with a posterior shift had a larger MME than knee OA without a posterior shift. Thus, this is the first study to clearly determine the relationship between the MME and the posterior shift of the tibia.

The contact area in the femorotibial joint increases significantly, and the stresses on the tibial cartilage are reduced by the meniscus [25]. The medial capsule was inserted into the medial meniscus border and moved forward with tibial anterior translation during knee extended [26]. Okazaki et al. [27] used MRI to demonstrate that the MME deviated anteromedially in the tibia. Furthermore, when the anterior translation of the tibia is restricted, the meniscus deviates anteromedially in the knee-extended position. When synovitis causes fibrosis and reduces the extensibility of the anterior synovial membrane and infrapatellar fat pad [28], it is inferred that the anterior segment of the medial meniscus is pulled anteromedially. In other words, we considered that the MME was greater in the posterior tibial translation group owing to the restricted anterior translation of the tibia with the meniscus in an anteromedial traction position.

Most treatments for MME have been reported in relation to surgical procedures, whereas few studies have provided evidence for conservative treatment. Yoshizuka et al. [29] demonstrated that static stretching of the semimembranosus muscle could reduce MME; however, the mechanism is not fully understood. Tsutsumi et al. [30] reported that the semimembranosus tendon was continuous with the posterior joint capsule and the medial meniscus was continuous with the joint capsule over the entire circumference. In addition, Tsutsumi et al. [31] reported that the joint capsule may dynamically coordinate the MM by transmitting semimembranosus action. In this study, we clarified the association between MME and posterior tibial translation. Therefore, the treatment of knee flexion contracture may be effective in improving MME.

This study had a few limitations. First, it was a cross-sectional study. Further intervention studies should be performed to clarify the relationship between MME and flexion contracture. Second, this study was conducted by a single therapist, and the results may depend on the examiner’s skill. In the future, it will be important to include data collected by multiple examiners. Third, age and height were not matched between the Early OA and Control groups. Fourth, knee alignment and MME were assessed without weight-bearing activity. The difference between weight-bearing and non-weight-bearing activity may be related to the progression of MME. Future studies should investigate the relationship between MME and alignment of the tibiofemoral joint during weight-bearing activity and walking. Fifth, it is also important to note that patients with severe deformities may not be suitable candidates for applying the methods utilized in this study. Finally, alignment in the frontal plane could not be assessed, but varus alignment is already known to be a factor in MME.

## 5. Conclusions

The results showed that the ultrasound imaging-based evaluation of tibial rotation alignment has high reproducibility, validity, and accuracy. Evaluating tibial rotational alignment using ultrasound imaging is a clinically straightforward and highly reliable method. Tibial rotation alignment was not found to be associated with MME, but tibial posterior shift alignment was. Treating knee flexion contractures may be effective in improving MME. Future studies should focus on the relationship between MME and alignment of the tibiofemoral joint during weight-bearing activities and intervention effects.

## Figures and Tables

**Figure 1 healthcare-12-02412-f001:**
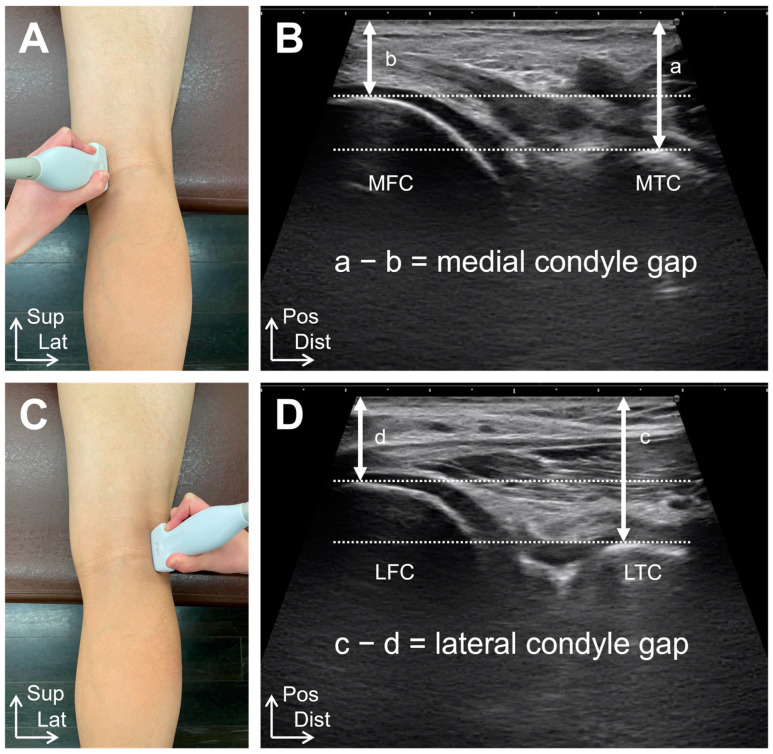
Ultrasound imaging measurement method. (**A**) Probe irradiation sites of medial femoral and tibial condyles. (**B**) Long-axis image of the most prominent posterior point of the medial femoral condyle and the most prominent posterior point of the medial tibial condyle. (**C**) Probe irradiation sites of lateral femoral and tibial condyles. (**D**) Long-axis image of the most prominent posterior point of the lateral femoral condyle and the most prominent posterior point of the lateral tibial condyle. a, the distance from the most prominent posterior medial tibial condyle to the skin. b, the distance from the most prominent posterior medial femoral condyle to the skin. c, the distance from the most prominent posterior lateral tibial condyle to the skin. d, the distance from the most prominent posterior lateral femoral condyle to the skin. MFC, medial femoral condyle; MTC, medial tibial condyle; LFC, lateral femoral condyle; LTC, lateral tibial condyle; Sup, superior; Lat, lateral; Pos, posterior; Dist, distal.

**Figure 2 healthcare-12-02412-f002:**
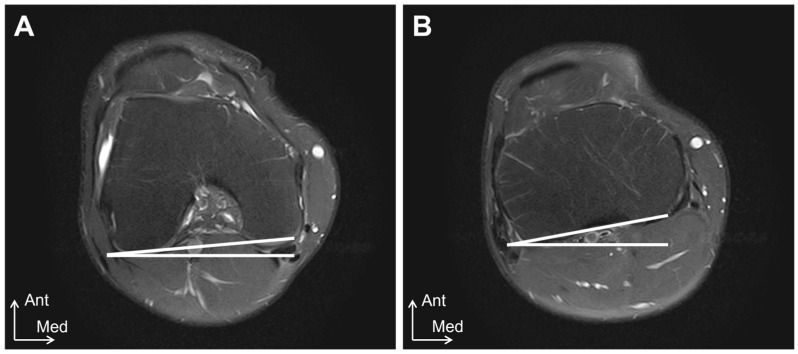
MRI tibial external-rotation angle measurement method. (**A**) femoral rotation angle. The femoral rotation angle was defined as the angle between the horizontal line and the line of the subchondral bone of the posterior femoral condyle at the most prominent point. (**B**) tibial rotation angle. The tibial rotation angle was defined as the angle between the horizontal line and the line of the posterior cortical margin of the tibial condyles using one slice above the head of the fibula. Ant, anterior; Med, medial.

**Figure 3 healthcare-12-02412-f003:**
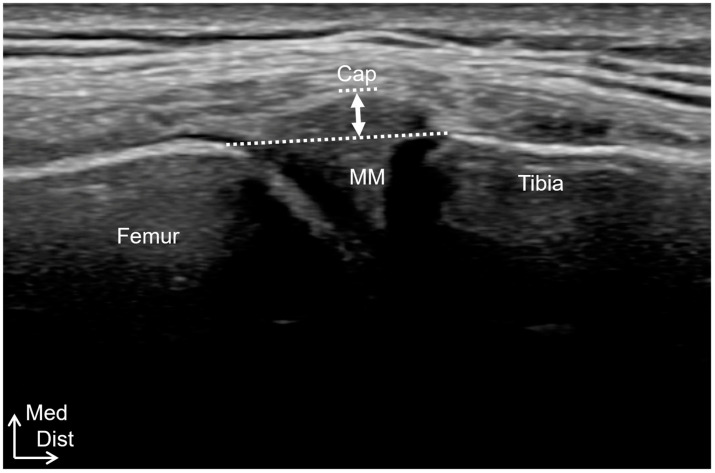
MME measurements. MME (white arrow) was measured as the vertical distance from the line segment connecting the medial end of the medial condyle of both the femur and tibia (white line), excluding the osteophytes, to the outer edge of the medial meniscus. MM, medial meniscus; Cap, joint capsule; Med, medial; Dist; distal.

**Figure 4 healthcare-12-02412-f004:**
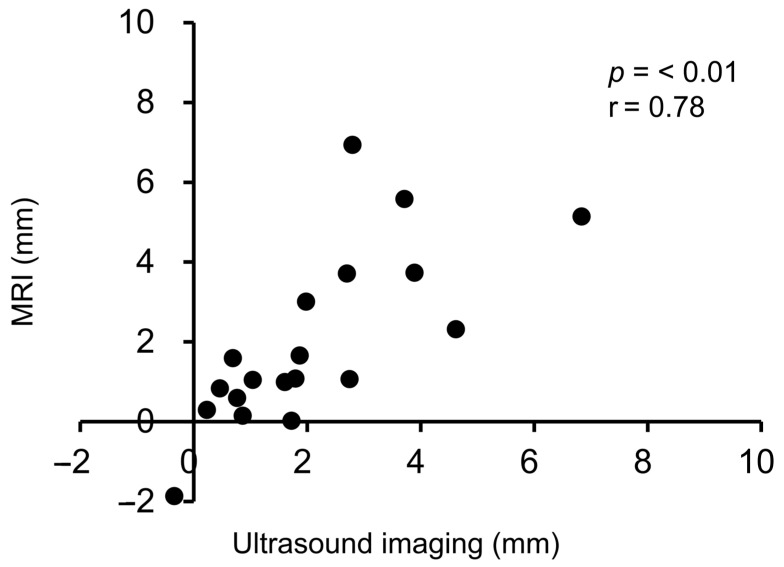
Correlation between the ultrasound imaging and the MRI ML gaps.

**Figure 5 healthcare-12-02412-f005:**
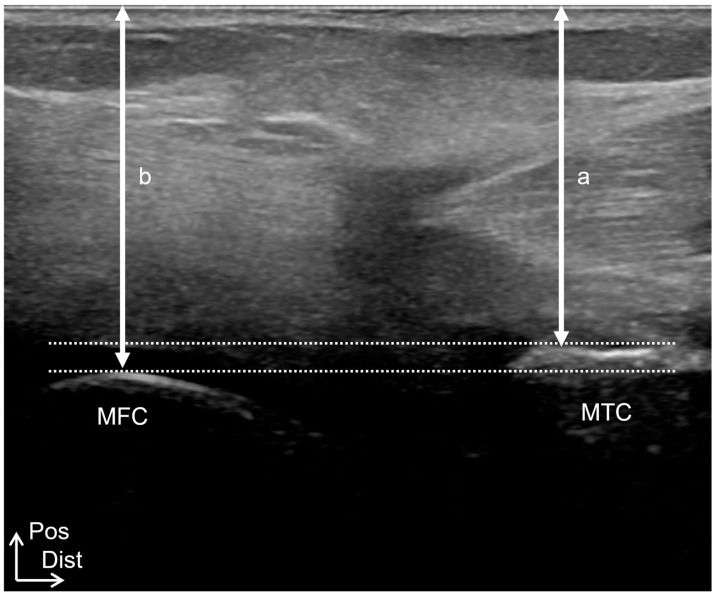
Knee OA with posterior shifted. A long-axis image was obtained to visualize the most prominent posterior points of the medial femoral condyle and the most prominent posterior point of the medial tibial condyles. a = 23.2 mm. b = 24.9 mm. Medial condyle gap (a − b) = −1.7 mm. MTC, medial tibial condyle; LFC, lateral femoral condyle; Pos, posterior; Dist, distal.

**Table 1 healthcare-12-02412-t001:** Intra- and inter-rater reliability.

Intra-Rater Reliability	ICC (1, 3)	95%CI	SEM	MDC95%
Distance from the medial condyle of the femur to the skin	0.973	0.901–0.993	0.565	1.566
Distance from the medial condyle of the tibia to the skin	0.979	0.922–0.995	0.598	1.658
Distance from the lateral condyle of the femur to the skin	0.881	0.615–0.969	1.026	2.844
Distance from the lateral condyle of the tibia to the skin	0.927	0.750–0.981	1.469	4.073
**Inter-Rater Reliability**	**ICC (2, 3)**	**95%CI**	**SEM**	**MDC95%**
Distance from the medial condyle of the femur to the skin	0.895	0.652–0.972	1.001	2.775
Distance from the medial condyle of the tibia to the skin	0.951	0.764–0.988	0.912	2.529
Distance from the lateral condyle of the femur to the skin	0.786	0.381–0.941	1.546	4.284
Distance from the lateral condyle of the tibia to the skin	0.859	0.369–0.966	2.052	5.687

ICC (1, 3), intra-rater correlation coefficient; ICC (2, 3), inter-rater correlation coefficient; CI, confidence interval; SEM, standard error of the mean; MDC95%, minimal detectable change at the 95% confidence interval.

**Table 2 healthcare-12-02412-t002:** Results of multiple regression analysis predicting MRI tibial rotation angle from the ML gap of ultrasound imaging and Q angle.

	Coefficient	β	SE	*t* Value	*p*-Value
Constant	1.732		0.962	1.790	0.09
ML gap	1.095	0.940	0.171	6.305	<0.01
Q angle	−0.049	−0.115	0.063	−0.774	0.45

SE: standard error. R^2^: 0.764. Variance inflation factor: 1.588.

**Table 3 healthcare-12-02412-t003:** Parameters of the Healthy, Early OA, and Moderate OA groups.

	Control (31 Knees)	Early OA (44 Knees)	Moderate OA (19 Knees)
	Mean ± SD	Mean ± SD	Mean ± SD
Age (years) ^†^	70.32 ± 6.49	64.64 ± 10.60	70.16 ± 8.77
Height (cm) ^†^	163.06 ± 5.67	157.84 ± 6.55	158.53 ± 6.18
Weight (kg)	57.42 ± 8.82	60.52 ± 9.58	62.16 ± 12.29
Knee extension angle (°) ^†§^*	−0.48 ± 1.50	−3.64 ± 3.12	−8.16 ± 3.42
Joint effusion (mm) ^†§^*	0.05 ± 0.13	1.37 ± 1.40	2.86 ± 1.90
Medial meniscus extrusion (mm) ^†^	2.54 ± 0.69	3.05 ± 0.80	3.04 ± 0.98
Medial Condyle gap (mm)	5.46 ± 1.67	6.03 ± 3.22	5.15 ± 4.30
Lateral Condyle gap (mm) ^§^	6.23 ± 1.59	6.80 ± 3.30	4.78 ± 2.90
Tibial external-rotation angle (°)	0.89 ± 2.49	2.18 ± 4.07	1.33 ± 5.44

^†^: Control-Early OA. ^§^: Early OA-Moderate OA. *: Control-Moderate OA.

**Table 4 healthcare-12-02412-t004:** Parameters for the group with normal, excessive external, and excessive internal rotations in knee osteoarthritis.

	Normal Rotation (34 Knees)	Excessive Internal Rotation (16 Knees)	Excessive External Rotation (13 Knees)
	Mean ± SD	Mean ± SD	Mean ± SD
Age (years)	64.29 ± 11.52	69.69 ± 8.23	67.38 ± 8.61
Height (cm)	157.47 ± 5.05	158.13 ± 8.04	159.46 ± 7.58
Weight (kg)	59.03 ± 7.61	63.13 ± 13.05	63.62 ± 12.71
Knee extension angle (°) ^†^	−3.82 ± 3.03	−6.56 ± 4.73	−6.15 ± 3.63
Joint effusion (mm) ^†^	1.38 ± 1.18	2.74 ± 2.18	1.84 ± 1.87
Medial meniscus extrusion (mm)	2.95 ± 0.85	3.06 ± 0.94	3.27 ± 0.74
Medial condyle gap (mm) ^†§^*	5.86 ± 2.36	8.20 ± 3.20	2.51 ± 4.25
Lateral condyle gap (mm) ^†§^	6.17 ± 2.55	4.41 ± 3.89	8.43 ± 3.11
Tibial external-rotation angle (°) ^†§^*	2.03 ± 1.32	−3.42 ± 2.24	8.21 ± 3.55

^†^: control-excessive internal rotation. ^§^: excessive internal rotation-excessive external rotation. *: control-excessive external rotation.

**Table 5 healthcare-12-02412-t005:** Parameters for the groups without and with a posterior shift in the knee osteoarthritis.

	Absence of Posterior Shift (39 Knees)	Presence of Posterior Shift (24 Knees)	*p*-Value	Effect Size
	Mean ± SD	Mean ± SD		
Age (years)	65.38 ± 10.43	66.87 ± 10.09	0.58	0.14
Height (cm)	158.29 ± 6.32	159.90 ± 6.36	0.81	0.06
Weight (kg)	64.04 ± 11.43	59.15 ± 9.07	0.07	0.99
Knee extension angle (°)	3.82 ± 2.77	−6.38 ± 4.41	<0.01	0.22
Joint effusion (mm)	1.31 ± 1.42	2.42 ± 1.82	<0.01	0.59
Medial meniscus extrusion (mm)	2.85 ± 0.73	3.28 ± 0.92	0.04	0.58
Medial Condyle gap (mm)	7.51 ± 2.37	3.71 ± 3.67	<0.01	0.76
Lateral Condyle gap (mm)	8.10 ± 2.69	3.94 ± 2.42	<0.01	1.05
Tibial external-rotation angle (°)	1.98 ± 3.42	1.83 ± 5.94	0.93	0.03

## Data Availability

The data are available upon request from the corresponding author.

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
