# Peer review of "Relationship Between the Sagittal and Coronary Alignment of the Tibiofemoral Joint and the Medial Meniscus Extrusion in Knee Osteoarthritis"

_healthcare, 2024, doi:10.3390/healthcare12232412_

Round 1
Reviewer 1 Report
Comments and Suggestions for Authors
The healthy control group consisted of only 10 individuals, which is very small. The small sample size may limit the generalizability of the results.
The age and physical characteristics of the osteoarthritis patients and the healthy individuals do not seem to be fully balanced. This may make it difficult to determine the true relationship between the variables.
The study was conducted with a cross-sectional design and only examined the current situation. This design is limited in revealing the causal relationship between the variables. A prospective design allows for more robust inferences about causality by observing changes over time.
Ultrasound assessment of the alignment of the tibiofemoral joint has been presented as a new method, but its accuracy has not been determined by comparing it with gold standard imaging methods (e.g. MRI or CT).
The measurements were made in a non-weight-bearing position. Weight-bearing measurements may provide more accurate results in assessing the progression of meniscus extrusion.
It does not seem that collinearity was controlled between the independent variables in the univariate and multiple regression analyses. This may lead to the omission of interactions between the variables.
It seems that multiple testing corrections were not applied when making comparisons between groups. Without multiple testing corrections, the probability of false positive results increases.
All ultrasound measurements were made by a single therapist, which may cause the measurements to be subjective. Blinding with independent observers provides more reliable results in terms of measurement accuracy and reproducibility.
Comments on the Quality of English Languageminor editing required
Reviewer 2 Report
Comments and Suggestions for Authors
I appreciate the opportunity to review this article and hope these comments contribute to enhancing its impact:
This manuscript looks into the relationship between the sagittal and coronal alignments of the tibiofemoral joint and MME in knee OA. The study has significant potential for better understanding how joint alignment impacts OA progression and treatment. However, while the research is quite interesting, the manuscript would benefit from some revisions to improve clarity, details in the methodology, and overall readability.
The introduction is well-done and provides a good background, but it would be stronger if it included more up-to-date references and a bit more discussion on why MME is important in OA. This would help place the study in a clearer context. Bringing in newer studies that show the current gaps in knowledge about tibiofemoral alignment would make the intro more impactful.
The methods section is detailed, but some parts could use clearer explanations. For instance, it’s not completely obvious why certain imaging methods were picked over others, like ultrasound versus MRI. Also, the participant selection criteria should be explained a bit more to make it transparent why certain people were included or excluded. Clarifying how variables were managed would add to the reliability of the methods.
The results section is thorough, but a few of the figures and tables could be more intuitive. Better labeling and more descriptive captions would make it easier for readers to quickly grasp what’s being shown. Simplifying the presentation and adding clearer summaries in the captions would help make the data more accessible.
The discussion does a solid job of interpreting the findings, but it could benefit from more comparisons with other similar studies. While the limitations are mentioned, it would be helpful if there was a bit more on how these might affect the study’s conclusions. Expanding on how these findings relate to other research and discussing their practical implications more deeply would make the discussion section stronger.
The conclusion does a good job summarizing the key points but could go further in suggesting how these findings could be applied in clinical practice or guide future research. More direct statements on potential applications would underline the importance of the results and show their value for future work.
Overall, the manuscript provides valuable insights into the role of knee joint alignment in OA. With these suggested changes, it would be clearer, better detailed, and more suitable for publication.
Round 2
Reviewer 1 Report
Comments and Suggestions for Authors
Article ready for publication.
Reviewer 2 Report
Comments and Suggestions for Authors
Thank you for the opportunity to review the revised manuscript. I appreciate the effort the authors have put into addressing the previous feedback in detail. This revised manuscript is well-written, methodologically sound, and provides valuable insights into knee OA. I am happy to recommend it for publication and commend the authors for their thoughtful revisions and rigorous work.